# Shotgun Analysis of Gut Microbiota with Body Composition and Lipid Characteristics in Crohn’s Disease

**DOI:** 10.3390/biomedicines12092100

**Published:** 2024-09-14

**Authors:** Péter Bacsur, Tamás Resál, Bernadett Farkas, Boldizsár Jójárt, Zoltán Gyuris, Gábor Jaksa, Lajos Pintér, Bertalan Takács, Sára Pál, Attila Gácser, Kata Judit Szántó, Mariann Rutka, Renáta Bor, Anna Fábián, Klaudia Farkas, József Maléth, Zoltán Szepes, Tamás Molnár, Anita Bálint

**Affiliations:** 1Department of Medicine, Albert Szent-Györgyi Medical School, University of Szeged, Kálvária Ave. 57, H-6725 Szeged, Hungary; 2HCEMM-USZ Translational Colorectal Research Group, H-6725 Szeged, Hungary; 3Momentum Epithelial Cell Signaling and Secretion Research Group, Hungarian Academy of Science, University of Szeged, H-6720 Szeged, Hungary; 4HCEMM-USZ Molecular Gastroenterology Research Group, H-6720 Szeged, Hungary; 5Delta Bio 2000 Ltd., H-6726 Szeged, Hungary; 6Mutagenesis and Carcinogenesis Research Group, Hungarian Centre of Excellence of Molecular Medicine, University of Szeged, H-6720 Szeged, Hungary; 7HCEMM-USZ Pathogen Fungi Research Group, H-6726 Szeged, Hungary

**Keywords:** IBD, inflammation, microbiota, obesity, lipid metabolism

## Abstract

Alterations to intestinal microbiota are assumed to occur in the pathogenesis of inflammatory bowel disease (IBD). This study aims to analyze the association of fecal microbiota composition, body composition, and lipid characteristics in patients with Crohn’s disease (CD). In our cross-sectional study, patients with CD were enrolled and blood and fecal samples were collected. Clinical and endoscopic disease activity and body composition were assessed and laboratory tests were made. Fecal bacterial composition was analyzed using the shotgun method. Microbiota alterations based on obesity, lipid parameters, and disease characteristics were analyzed. In this study, 27 patients with CD were analyzed, of which 37.0% were obese based on visceral fat area (VFA). Beta diversities were higher in non-obese patients (*p* < 0.001), but relative abundances did not differ. *C. innocuum* had a higher abundance at a high cholesterol level than Bacillota (*p* = 0.001, *p* = 0.0034). *Adlercreutzia*, *B. longum*, and *Blautia* alterations were correlated with triglyceride levels. Higher Clostridia (*p* = 0.009) and *B. schinkii* (*p* = 0.032) and lower *Lactobacillus* (*p* = 0.035) were connected to high VFA. Disease activity was coupled with dysbiotic elements. Microbiota alterations in obesity highlight the importance of gut microbiota in diseases with a similar inflammatory background and project therapeutic options.

## 1. Introduction

The human body is inhabited by various microbes, forming a complex ecological system that plays a fundamental role in health and disease [1]. Numerous studies have explored the physiological and pathological aspects of human microbiota [2]. The development of molecular techniques for microbial analysis provides an opportunity to uncover more precise connections between microbes and their human hosts, leading to a better understanding of physiology and potential therapeutic implications [3].

Chronic disorders are often characterized by alteration of the gut microbiota, which are believed to be involved in disease pathogenesis [4]. In obesity, several pathological connections have been raised according to gut microbiota changes. An increase in the Bacillota/Bacteroidota ratio was published, while decreased diversity was reported as well [5,6,7]. The microbiota-modulating effect of nutritional habits is clear in obesity, and the impact of the microbiota in increasing the risk for obesity has also been raised, on the other hand, since translational studies revealed that gut bacteria can translocate into the submucosal adipose tissue, promoting adipogenesis [8]. Obesity can be defined as a pro-inflammatory state, since adipose tissue is responsible for the secretion of a number of pro- and anti-inflammatory cytokines [9].

In IBD, decreased diversity and an altered Bacillota/Bacteroidota ratio was also seen [5]. The link between microbiota change and IBD has become univocal over the last decade, but not all of the details are clear. Overall, a decrease in the diversity of the intestinal microbiota is a key factor influencing the course of IBD, which is in line with the data published on obesity [6,9].

A clear shift in intestinal microbiota and decreased diversity result in epithelial barrier dysfunction [4]. Throughout the leaky epithelial barrier, bacterial antigen presentation is initiated, which creates and maintains low-grade systemic inflammation [9]. The role of this key process is also supported in IBD by several reports, while it may create a link between colorectal carcinoma, metabolic syndrome, obesity, diabetes, and other diseases associated with systemic inflammation [10,11].

Metabolic dysfunction of lipid homeostasis is often coupled with obesity, which raises the possible similar background of different pathological conditions and the impact of low-grade systemic inflammation. Furthermore, an altered Bacillota/Bacteroidota ratio was found [12], while a decreased abundance of short-chain fatty-acid-producing species was found in patients with dyslipidaemia. This latter finding clearly increases the risk of developing leaky gut [13]. There is evidence that an increased level of adiponectines and leptines promotes inflammation, while the pro-inflammatory cytokine TNFα leads to insulin resistance by inhibiting tyrosine kinase activity, which results in a vicious circle affecting cholesterol and triglyceride regulation [12,13].

A recent study revealed that IBD- and (BMI-based) obesity-specific species and pathways potentially play important roles in regulating the gut microbial balance; however, neither the lipid metabolism nor the nutritional parameters of the included patients were taken into account [14,15].

Based on these results, the importance of intestinal microbiota in the pathogenesis of IBD and metabolic dysfunction, especially in overweight or obese subjects, is hypothesized; however, evidence is still being gathered. We hypothesized that changes in the intestinal bacterial composition of patients with IBD are associated with metabolic dysfunction, especially of the lipid metabolism, and consequential obesity.

## 2. Materials and Methods

### 2.1. Patient Recruitment

In this cross-sectional study, consecutive patients with CD with different disease extensions and behaviors treated at the Department of Medicine, University of Szeged, were enrolled. They were assessed based on disease phenotype, clinical and endoscopic activities, and type of therapy received. The patient enrollment period was from January 2019 to March 2020.

All of the patients who agreed to participate in the study signed an informed consent form. The inclusion criterion was definitive CD for at least 3 months before enrollment. Exclusion criteria were age below 18 years, recent gastrointestinal surgery (within 3 months of study recruitment), antibiotic or probiotic use 6 weeks before inclusion, steroid titration within 6 weeks prior to study entry, chronic NSAID use or NSAIDS within 6 weeks of study recruitment (apart from 5-ASA therapy), continuous PPI use within 6 weeks prior to involvement, dysphagia, pregnancy, inability to give informed consent, severe concomitant illness, and active infectious disease.

### 2.2. Data Collection

Baseline demographic information (age, gender, weight, and height) and clinical data, such as disease duration and classification using Montreal definitions [16], concomitant treatments, and prior surgeries, were recorded for all patients. Blood and fecal samples were collected during the baseline visit for biomarker determination, which included C-reactive protein (CRP) level, leukocyte count and thrombocyte count, serum iron, hematocrit, hemoglobin, fecal calprotectin, and fecal microbiota. Fecal calprotectin levels were measured using the ELISA method from ORGENTEC^©^ Diagnostika GmbH in Mainz, Germany. Disease activity was assessed using the Crohn’s Disease Activity Index (CDAI) [17]. As a gold standard, colonoscopy was performed within 2 weeks of inclusion by trained gastroenterologists with extensive experience in optical colonoscopy, following established guidelines. The colonoscopy was used to assess the location and severity of mucosal inflammation. Conscious sedation was administered if requested by the subject.

Concomitant treatments continued during the study period, such as 5-aminosalicylate, corticosteroid, immunosuppressants, and/or biological therapy, were assessed.

### 2.3. Outcome Measurements and Definitions

The associations between parameters describing high visceral fat and overweight/obesity (visceral fat area, VFA; body mass index, BMI; waist–hip ratio, and results of body composition analysis) and gut microbiota were assessed as primary endpoints. The association of intestinal microbiota alterations and serum lipid metabolism markers (cholesterol and triglyceride) were analyzed as secondary endpoints. Tertiary endpoints were microbiota alterations connected to disease activity, phenotype, and treatment of CD.

Obesity was primarily defined using VFA (>100 cm^2^). A CDAI score above 150 was considered to be indicative of clinical activity, while endoscopic activity was determined by the disease-specific endoscopic score (Simplified Endoscopic Activity Score for Crohn’s disease-SES-CD) [18], and activity was determined by score of >2.

### 2.4. Body Composition Analysis

An InBody770^®^ device was used in the study to perform simultaneous multi-frequency bioelectrical impedance measurements. This allowed for the determination of various body composition parameters, including total body water volume, individual water spaces, body fat mass, dry mass, and muscle mass. The device treats the body as five cylinders to ensure accurate measurements. It also provides information on body composition components such as total water mass, body fat, protein, and minerals. Additionally, the InBody770^®^ assesses visceral fat area, muscle–fat analysis, visceral fat analysis (BMI, body fat percentage), segmented soft tissue analysis, and segmental fat analysis. It also measures extracellular and intracellular water, basal metabolic rate, and waist–hip ratio.

Visceral fat was measured via body composition analysis and was used to categorize patients into obese (above 100 cm^2^) and non-obese groups [19]. VFA was calculated based on the amount of fat in abdominal organs and their surroundings. Obesity, according to the WHO definition, is a condition caused by elevated fat tissue associated with an increased risk of disease. Elevated fat tissue can be measured using BMI, calculated using the weight/height^2^ formulation, and BMI above 25 kg/m^2^ was classified as the cut-off value of high visceral fat (overweight and obesity) [20].

### 2.5. Nutritional Questionnaire

Since there is no validated questionnaire available on this topic in the Hungarian language, a detailed questionnaire was developed, which includes lifestyle characteristics, dietary habits, and the features and uses of dietary supplements. A nutritional questionnaire was used to exclude patients based on strict exclusion criteria.

### 2.6. Determination of Microbiome Component

Stool samples were obtained and included in 8 mL plastic tubes (Biolab^®^, Budapest, Hungary) without buffer and stored at −20 °C until further determination of the microbiome components. Measurements were performed within two weeks after sample collection.

The bacterial DNA from the stool samples was extracted using a ZR Fecal DNA MiniPrep™ kit (Zymo Research, Irvine, CA, USA), following the manufacturer’s instructions, which included bead-beating mechanical lysis. After isolation, DNA samples were stored at −80 °C until shotgun sequencing.

For library preparation, the Illumina NexteraXT kit (Illumina Inc., San Diego, CA, USA) was used following the manufacturer’s instructions. Prior to library preparation, the genomic DNA from the fecal samples were quantified using a DNA High-Sensitivity kit (Agilent Technologies, Santa Clara, CA, USA) on a Qubit instrument. During library preparation, the DNA was fragmented using tagmentation, which also involved tagging the samples with unique indices. The size and purity of the completed libraries were assessed using a BioAnalyzer 2100 with a High-Sensitivity DNA kit. For sequencing, approximately 1 million reads were run per sample.

Identification of bacterial species from the shotgun sequences was performed using a custom alignment-based method [21]. We constructed a highly reliable bacterial and archaeal database utilizing representative and reference genomes from NCBI RefSeq [22]. In order to avoid ambiguous identification, we selected one reference genome per species.

Shotgun sequences were processed using Trimmomatic and Cutadapt with default settings in order to remove adapter sequences and low-quality bases [23,24]. The BWA algorithm was then used to align the reads to the reference database. Alignments were filtered in order to avoid false-positive species; not sufficiently covered genomes were excluded, and the relative abundance of bacteria was calculated based on the remaining reads.

### 2.7. Statistical Analysis

Statistical analysis was conducted using IBM SPSS (IBM SPSS Statistics for Windows, Version 25.0, IBM Corp., New York, NY, USA) and R software (version 4.1.1; R Foundation, Vienna, Austria). The normality of the data was assessed using the Shapiro–Wilk test. However, considering the proportions of each taxon in the human population, non-normality is considered to be almost baseline. Descriptive statistics were reported as count and percentages for categorical variables, and as mean ± standard deviation of the mean and median + interquartile range (IQR) for continuous variables. As the datasets had a relatively small number of items and lacked normality, Mann–Whitney and Kruskal–Wallis tests were used to compare groups with continuous variables. For continuous variables, a Spearman correlation was performed to explore the relationship between all taxa and the explanatory variable of interest. Categorical variables were compared using chi-squared tests and Fisher’s exact tests. A *p*-value of less than 0.05 was considered to be statistically significant. The Bonferroni correction was used to reduce multiple comparisons bias.

The Shannon diversity index was calculated to compare microbiota profiles based on the alpha diversity, which measures the bacterial richness of a population [25]. A beta diversity calculation was performed to compare bacterial diversity at the beta level between samples [26]. To calculate the Shannon index and Bray–Curtis distance, statistical comparisons were performed between the normal and obese groups, and the distances were visualized using Python (version 3.7). Pairwise distances were calculated for each sample, and the distances were averaged for the groups (normal and obese) using math and statistics packages. The Kruskal–Wallis test was employed to determine the significance of the difference between groups using the statannot package. Visualization was performed using the matplotlib and seaborn packages.

### 2.8. Ethical Approval

The study protocol was approved by the Scientific Research Ethics Committee of the Hungarian Medical Research Council’s proposal (52/2019-SZTE). This study was conducted according to the principles of the Declaration of Helsinki (1975 Declaration of Helsinki, 6th revision, 2008). Patients provided signed informed consent for study participation prior to enrollment.

## 3. Results

### 3.1. Baseline Characteristics of the Study Population

In the study, 27 patients with CD were investigated after applying the inclusion and exclusion criteria. The male-to-female ratio was 0.33, and the median age of the cohort was 35 (26–40) years. Ileocolonic and colonic localizations were equally observed in 11/27 (40.7%) cases, while the luminal phenotype was present in 14/27 (51.9%) of patients. Clinical activity and CRP levels indicated mild elevation, and the majority of patients had moderate endoscopic activity based on the SES-CD score. Biological treatment was received by 16/27 (59.3%) of patients, and 8/27 (29.6%) were on thiopurine therapy. Regarding VFA and BMI, 10/27 (37.0%) and 12/27 (44.4%) of patients were classified with high visceral fat, respectively. Differences between obese and non-obese groups were not identified according to baseline descriptive data. More detailed baseline demographic and clinical data can be found in Table 1. Baseline data from the body composition analysis of enrolled patients can be found in Table 2.

### 3.2. Association between Parameters Describing Obesity, Crohn’s Disease Phenotype, Prognosis, and Microbiome

The relative abundancies of obese and nonobese patients based on VFA did not differ; however, Bray–Curtis distances between samples were higher in the non-obese cohort (Figure 1, *p* < 0.001). A principal coordinate analysis showed separated samples (Figure 2).

The analysis showed increased abundances of class Clostridia (Figure 3, *p* = 0.009), with *Blautia schinkii* (*p* = 0.032, ρ = 0.421) amongst patients with higher VFA.

The abundance of *Adlercreutzia equolifaciens* was nearly zero above 100 cm^2^ VFA, and this correlation was confirmed using a significant ANOVA analysis (*p* = 0.027). Similar inverse proportionality was observed for the order *Lactobacillales* (*p* = 0.007, ρ = −0.515) and genus Lactobacillus (*p* = 0.01, ρ = −0.493) associated with VFA.

According to the WHO definition of overweight and obesity, elevated BMI was also examined in association with CD. Two subjects with lower than normal BMI were excluded from the analysis. A Mann–Whitney analysis revealed a significant difference between groups in relation to the abundance of *Cutibacterium* acnes (*p* = 0.041) and a nearly significant difference with *Adlercreutzia equolifaciens* (*p* = 0.055). Higher BMI was associated with greater abundance of *Cutibacterium* acnes and lower abundance of *Adlercreutzia equolifaciens*. Lower body weight (in kilograms) was correlated with a higher richness of *Bifidobacterium bifidum* species (*p* = 0.01, ρ = −0.493). Statistical analyses indicated changes in numerous bacterial families and species associated with waist–hip ratio as a descriptive character of obesity. Notably, the abundance of *Clostridia* (*p* = 0.04, ρ = 0.405), *Eubacteriales* (*p* = 0.04, ρ = 0.405), and *Roseburia hominis* (*p* = 0.048, ρ = 0.391) showed equal correlations with waist–hip ratio.

Body fat mass, including subcutaneous and visceral fat, increased with the abundance of *Blautia schinkii* (*p* = 0.021, ρ = 0.451) and decreased in parallel with *Lactobacillus paragasseri* (*p* = 0.015, ρ = −0.470) and *Roseburia intestinalis* (*p* = 0.035, ρ = 0.414). On the other hand, fat-free mass showed an inverse correlation with *Bifidobacterium bifidum*, which had a significantly higher proportion in individuals with lower fat-free mass (*p* = 0.029, ρ = −0.428). A similar result was observed when examining differences in gut flora for skeletal muscle only, with higher abundance of *Bifidobacterium bifidum* in patients with low SMM (*p* = 0.029, ρ = −0.428). Diet significantly affected the abundance of the genus *Blautia* (*p* = 0.039). However, due to the small number of subgroups, it is not possible to determine which diet had the greatest effect. About 73.3% of patients followed a diet low in fat, fiber, and lactose. Among the subjects with a low fat–fiber–lactose diet, 5 out of 15 had a high BMI.

Anatomical status also had an impact on the intestinal microbiome. Patients who underwent resection had a significantly decreased abundance of the orders *Bacteroidales* (Figure 4, *p* = 0.021) and *Eubacteriales* (*p* = 0.028). Significant changes were observed in *Roseburia* (*p* = 0.004). However, there was no difference in the presence of high visceral fat (based on BMI) in relation to resection or intensified therapies (corticosteroids or biological therapy). The B2 (stricturing) and B3 (penetrating) phenotypes of CD were more frequent (*p* = 0.036), and the B2 phenotype was associated with a higher abundance of *Bacteroides mediterraneensis* (*p* = 0.013) and *Lachnospira eligens* (*p* = 0.01). Further results of body composition analyses did not reveal these findings.

### 3.3. Association between Lipid Metabolism and Microbiome

Triglyceride and cholesterol levels were analyzed as serum markers of lipid metabolism. Serum cholesterol showed a positive correlation with the *Bacteroidota* strain, especially *Clostridium innocuum* (*p* = 0.034, ρ = 0.434), and an inverse correlation with the abundance of *Bacillota* (*p* = 0.001, ρ = −0.618), as demonstrated with a Spearman correlation analysis. Higher *C. innocuum* abundance (*p* = 0.011) was found in the presence of chronic metabolic diseases (diabetes mellitus, hypertension, etc.), although the interpretation of this result is limited by the unbiased distribution of the groups (8 patients with metabolic abnormalities and 18 without abnormalities). Serum triglyceride levels were associated with a significantly higher amount of *Blautia hanseni* (*p* = 0.045, ρ = 0.413) and decreased abundance of *Adlercreutzia equolifaciens* (*p* = 0.037, ρ = −0.427) and *Bifidobacterium longum* (*p* = 0.006, ρ = −0.545).

Triglyceride and cholesterol levels did not differ between obese and non-obese groups (based on VFA), and VFA did not show any correlation with serum lipid metabolism markers.

### 3.4. Association between Crohn’s Disease Activity and Microbiome

Clinical, biochemical, and endoscopic activity was associated with dysbiotic elements, and thus reduced abundance of certain species were identified, e.g., *Blautia obeum* and *Roseburia hominis*. A significant relationship between *Blautia* and *Roseburia* species and activity markers are shown in Table 3. Other significant connections with different microbiota alterations were not found.

### 3.5. Association between Crohn’s Disease Treatment and Microbiome

We considered the interplay between corticosteroids, azathioprine, and biological therapy and the microbiome to investigate the impact of immunosuppressive therapy. During systemic corticostreoid therapy, the abundance of the *Pasteurellacea* family was lower (*p* = 0.005). Azathoprin was associated with a decreased amount of *Clostridium leptum* (*p* = 0.013) and an increased amount of *Blautia wexlerae* (*p* = 0.039). Patients with biological therapy showed a higher abundance of *Roseburia inulinivorans* (*p* = 0.02) and lower *Bifidobacterium bifidum* (*p* = 0.005). Seven of ten (70%) patients elevated VFA, and eight of twelve (66.6%) patients with high BMI needed biological therapy.

## 4. Discussion

The possible connection between lipid metabolism and body composition-associated microbiota alterations in patients with CD was investigated. There were higher Bacteroidota and *C. innocuum* and lower Bacillota abundances at high serum cholesterol levels. The richness of *B. hanseni* and decreased abundance of *B. longum* and *A. equolifaciens* were associated with high serum triglyceride levels. High visceral fat based on VFA, BMI, and body weight was connected to an increased abundance of *Cutibacterium acnes* and *B. schinkii* and a decreased abundance of *A. equolifaciens*, *B. bifidum*, *Lactobacillus*, and *Roseburia*. CD activity was coupled with dysbiotic *Blautia* and *Roseburia* elements. Higher *Roseburia* and lower *Bifidobacterium* levels were seen during the biological treatment.

An increasing number of IBD patients characterized by obesity are presenting at regular visits. The data suggest that high visceral fat and obesity are associated with increased disease activity, development of extraintestinal symptoms, reduced efficacy of biological treatments, and poor disease outcomes [27,28].

The possible association between obesity and gut microbiota has been raised in recent years [28]. Ha et al. showed the tendency of bacterial translocation via the epithelial barrier into the submucous adipose tissue. *C. innocuum* stimulated tissue remodeling and promoted adipogenesis, which was confirmed in gnotobiotic mice [8]. It is known that adipose tissue is responsible for the secretion of a number of pro-inflammatory cytokines via adiponectines and leptines (TNFα, interleukine 6) which maintain a low grade of inflammation. An elevated level of TNFα inhibits tyrosine kinase activity and results in insulin resistance, which has an impact on worsening metabolic homeostasis (diabetes and dyslipidaemia) and, consequently, worse IBD symptoms [9]. A Ukrainian population-based study found a positive association between the Bacillota/Bacteroidota ratio and BMI [29], which highlights the importance of altered microbiota on disease pathogenesis [30]. *B. longum* was investigated for its potential use in a therapeutic approach for obesity by modifying adipogenesis in a mouse model [31,32], suggesting the existence of a role for Bifidobacteria in the development of obesity. Recent research has presented alterations of Lactobacillus in obesity [33]. Although data at the species level contradict each other, multiple papers suggest the effectiveness of Lactobacillus-based probiotics in the treatment of obesity and IBD by increasing the Bacillota/Bacteroidota ratio [5]. The exact pathogenetic connections regarding specific microbiota alterations are still not available, but publications suggest that decreased diversity and shifts in microbial composition can contribute to the development and worsening of obesity, which is a key point in the background of IBD [6,7].

In this study, we identified an increased abundance of Clostridia in patients with high VFA, particularly *B. schinkii*, while a high BMI level was associated with a richness of *Cutibacterium acnes* and reduced abundance of *Adlercreutzia equolifaciens*. An inverse correlation was observed between Lactobacillales and Lactobacillus abundances with VFA. *B. schinkii* was connected with body fat mass, and an inverse correlation was found between body fat mass and *Roseburia intestinalis* and *L. paragasseri*.

Gut microbiota alterations were revealed in connection with serum lipid parameters. A higher serum cholesterol level was characterized with decreased Bacillota/Bacteroidota ratio and increased abundance of *C. innocuum* which highlight the importance of altered gut microbiota and decreased diversity in pathological conditions. Correlations between serum triglyceride levels and Adlercreutzia, Blautia, and Bifidobacterium species were detected in our cohort. A possible explanation for the causative connection between lipid parameters and microbiota alterations is presented in experimental studies. Jing et al. reported that Fubrick tea aqueous extract would increase the relative abundances of Adlercreutzia, Streptococcus, and Bacteroides and help regulate lipid metabolism by short-chain fatty acid (SFCA) production [34]. SFCA production by intestinal fiber fermentation could promote mucosal healing and intact tight junction formation in the intestinal epithelial barrier. Conversely, when the epithelial barrier is weakened, the mucosal layer of the gut becomes permeable, and antigen presentation can maintain a low grade of inflammation. Blautia and Roseburia species modulate the native immune response and help to reduce mucosal inflammation via SFCA production [35,36]. High fat intake and a high level of serum lipids make the gut permeable due to oxidative stress, so bacterial toxins can also contribute to the development of a pro-inflammatory state, which connects obesity, dyslipidaemia, and IBD [37]. Pro-inflammatory predominance can worsen IBD symptoms in addition to exacerbating existing metabolic dysfunction [27]. Our analysis showed a mild-to-moderate correlation between disease activity and Blautia and Roseburia abundances.

Several studies have investigated the association between different therapies and gut microbiota in IBD. In our study, alterations in Blautia and Roseburia were associated with the activity markers of CD. Lower levels of *F. prausnitzii*, Bifidobacterium, and various Clostridium species were associated with increased clinical activity in patients with ulcerative colitis (UC) [37,38]. A pediatric study on IBD found that a higher abundance of Bifidobacterium could predict responses to anti-TNF therapy, while decreased baseline diversity was associated with an increased risk of anti-TNF failure [39]. There microbiota changes may also develop in association with obesity, while obesity alone is also associated with treatment failure. As a result, obesity in IBD clearly increases the risk of disease progression and treatment failure. In a study on UC, a higher abundance of Roseburia was found in cases of successful anti-integrin therapy [40]. In our study, a lower abundance of Bifidobacterium and a higher abundance of Roseburia were associated with the use of biological treatments. It should be highlighted that the gut microbiota composition fundamentally influences the bioavailability of several oral drugs [41]. Therefore, the decreased efficacy of oral 5-ASA, corticosteroids, or thiopurines may be explained by altered intestinal microbiota.

Several factors limit the interpretation of our current study. First, the total number of participants is below the ideal number for drawing precise conclusions. Longitudinal sample measurements could have provided more refined results, but the cost of shotgun sequencing compared to the 16S rRNA method made it impractical in our study. Moreover, the design of our analysis did not allow us to differentiate between luminal and mucosal microbial changes. We also did not analyze viral and fungal components of the gut microbiota, nor did we assess the metabolites produced by intestinal organisms. It is important to note that our study focused on the microbiota characteristics of patients with CD, so we cannot draw an exact causative relationship with microbiota alterations due to the selected population. It should be highlighted that our study’s cross-sectional design does not allow us to examine the associations between IBD, obesity, and the dysfunction of lipid metabolism per couple. Furthermore, gut microbiota is influenced by various environmental factors, and despite our best efforts to control these, we may not have achieved full control over all environmental variables. In this manner, nutritional habits were not included in further statistical analyses due to the low number of enrolled participants.

Despite the limitations mentioned above, our study contributes to the limited literature available on the connection between microbiota composition and obesity in the selected IBD population. The strength of our analysis lies in the use of shotgun sequencing, which provides more accurate results compared to the 16S rRNA sequencing method. Simple comparisons also minimize the risk of type I errors compared to multiple testing.

## 5. Conclusions

In conclusion, our study identified significant associations between lipid metabolism and obesity characteristics and microbiota alterations in patients with CD. The role of increased abundance of *Blautia* and *Clostridia* and decreased abundance of *Bifidobacteria*, *Adlercreutzia*, and *Lactobacillus* may be highlighted, as it helps us to understand the etiology of the disease and promotes the development of personalized treatments, such as microbiota-modifying agents. Addressing obesity in patients with CD may be necessary to modify the associated microbial alterations and achieve better therapeutic responses and disease control. A multidisciplinary approach involving dietitians in IBD management is required.

## Figures and Tables

**Figure 1 biomedicines-12-02100-f001:**
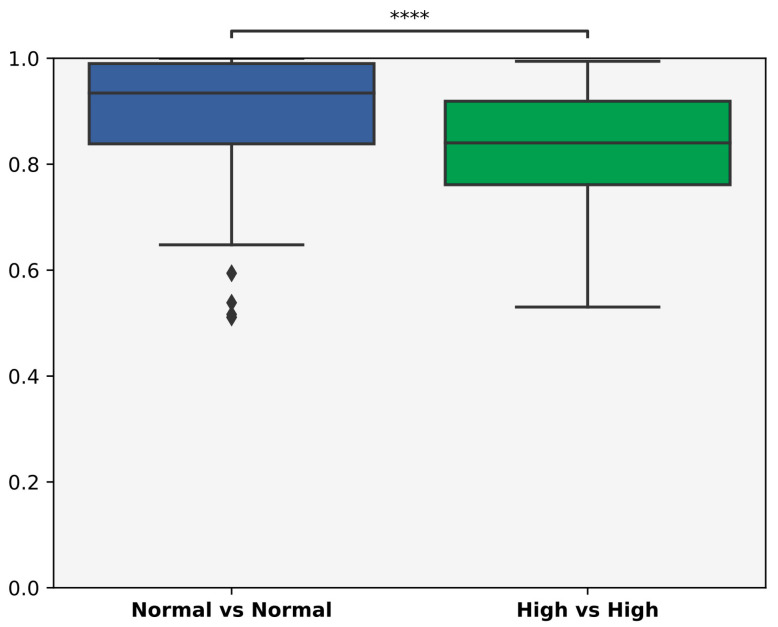
Bray–Curtis distances between samples in the obese vs. nonobese groups using visceral fat area as a grouping factor. The relative abundances of obese and nonobese patients (based on visceral fat area) did not differ between cohorts. However, non-obese participants had significantly higher distances (****: *p* > 0.001).

**Figure 2 biomedicines-12-02100-f002:**
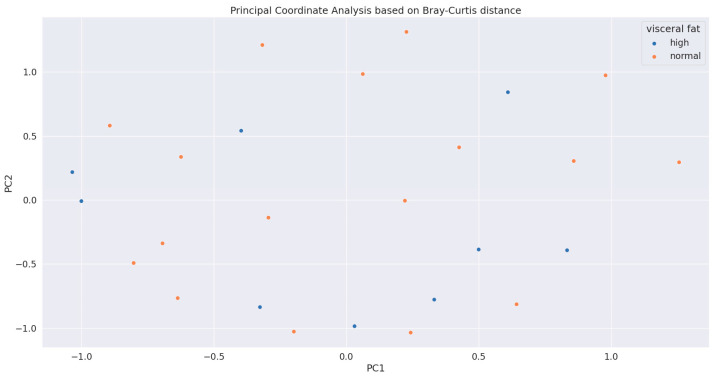
Principal coordinate analysis of obese and non-obese samples (based on visceral fat area) showed separated dots.

**Figure 3 biomedicines-12-02100-f003:**
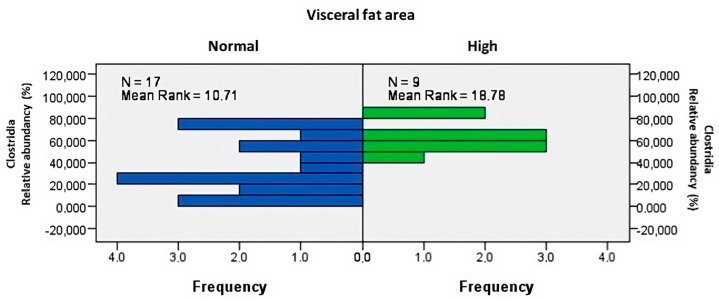
Higher visceral fat area was associated with increased abundances of class *Clostridia* (*p* = 0.009).

**Figure 4 biomedicines-12-02100-f004:**
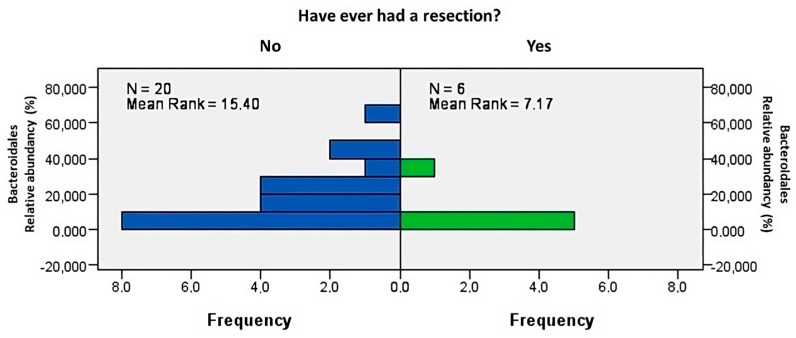
Prior intestinal resection was associated with decreased abundance of *Bacteroidales* (*p* = 0.021).

**Table 1 biomedicines-12-02100-t001:** Baseline demographic and clinical characteristics of the enrolled participants.

Characteristics	Total (N = 27)	Obese * (N = 10)	Non-Obese * (N = 17)	Sig.
Gender (male, N, %)	9 (33.3)	4 (40.0)	5 (29.4)	0.573
Age (years, median (IQR))	35 (26–40)	36 (29–53)	35 (25–40)	0.070
Disease duration (years, median (IQR))	7 (2–13)	9 (5–18)	4 (1–13)	0.251
CD localization (N, %) ^+^				
ileal	5 (18.5)	0 (0)	5 (29.4)	0.111
colonic	11 (40.7)	4 (40.0)	7 (41.2)	
ileocolonic	11 (40.7)	6 (60.0)	5 (29.4)	
upper gastrointestinal disease	2 (7.4)	0 (0)	2 (11.8)	0.516
CD behavior (N, %) ^+^				
non-stricturing, non-penetrating	14 (51.9)	3 (30.0)	11 (64.7)	0.157
stricturing	5 (18.5)	2 (20.0)	3 (17.6)	
penetrating	8 (29.6)	5 (50.0)	3 (17.6)	
Previous bowel resection (N, %)	6 (22.2)	1 (10.0)	5 (29.4)	0.363
Disease activity				
CDAI at inclusion (mean, ± SD)	146.2 (98.8)	112.4 (113.6)	164.3 (88.7)	0.165
SES-CD at inclusion (mean, ± SD)	7.0 (7.9)	7.9 (10.6)	6.5 (6.4)	0.948
Fecal calprotectin (ug/g, median, [IQR])	523.7 (327.9)	552.3 (377.5)	508.4 (311.5)	0.532
CRP at inclusion (mean, ± SD)	8.5 (8.5)	12.1 (11.6)	6.6 (6.4)	0.345
Therapy (N, %)				
5-ASA	6 (22.2)	3 (30.0)	3 (17.6)	0.638
oral corticosteroid	5 (18.52)	2 (20.0)	3 (17.6)	1.000
topical corticosteroid	1 (3.7)	1 (10.0)	0 (0)	0.370
thiopurines	8 (29.6)	4 (40.0)	4 (23.5)	0.415
biological therapy	16 (59.3)	7 (70.0)	9 (52.9)	0.448
infliximab	4 (14.8)	2 (20.0)	2 (11.8)	-
adalimumab	4 (14.8)	1 (10.0)	3 (17.6)	
vedolizumab	4 (14.8)	1 (10.0)	3 (17.6)	
ustekinumab	4 (14.8)	3 (30.0)	1 (5.9)	
Obesity characteristics				
VFA > 100 cm^2^ (N, %)	10 (37.0)	-	-	-
BMI > 25 kg/m^2^ (N, %)	12 (44.4)	-	-	-
Serum cholesterol (mmol/L, mean, ± SD)	4.3 (0.9)	4.5 (0.8)	4.2 (0.9)	0.421
Serum triglyceride (mmol/L, mean, ± SD)	1.5 (1.1)	1.9 (1.1)	1.4 (1.0)	0.217

Abbreviations: N: number of patients, IQR: interquartile range, CD: Crohn’s disease, SD: standard deviation of mean, CDAI: Crohn’s disease activity index, SES-CD: Simple Endoscopic Score for Crohn’s disease, CRP: C-reactive protein, 5-ASA: 5-aminosalicylate, VFA: visceral fat area, BMI: body mass index, Sig: significance level, * Based on visceral fat area VFA > 100 cm^2^, ^+^ based on Montreal classification.

**Table 2 biomedicines-12-02100-t002:** Body composition parameters of included patients.

Variable	Total (N = 27)	Obese * (N = 10)	Non-Obese * (N = 17)	Sig.
Body weight (kg, mean, ± SD)	72.2 (18.9)	88.0 (16.4)	62.9 (13.5)	**<0.001**
Body mass index (kg/m^2^, mean, ± SD)	24.8 (5.2)	29.7 (4.6)	21.9 (2.9)	**<0.001**
Total body water (L, mean, ± SD)	37.8 (8.6)	41.3 (8.0)	35.7 (8.5)	0.051
Soft lean mass (kg, mean, ± SD)	48.5 (11.0)	53.0 (10.2)	45.8 (10.9)	0.051
Fat-free mass (kg, mean, ± SD)	51.5 (11.7)	56.2 (10.9)	48.7 (11.6)	0.054
Body fat mass (kg, mean, ± SD)	20.87 (11.1)	31.8 (10.0)	14.5 (5.1)	**<0.001**
Skeletal muscle mass (kg, mean, ± SD)	28.4 (6.9)	31.2 (6.4)	26.7 (6.8)	0.050
Percent body fat (%, mean, ± SD)	27.7 (9.0)	35.8 (7.1)	22.8 (6.0)	**<0.001**
Visceral fat area (cm^2^, mean, ± SD)	95.1 (52.1)	149.9 (43.2)	62.9 (19.6)	**<0.001**
Waist–hip ratio (mean, ± SD)	0.9 (0.08)	1.0 (0.03)	0.85 (0.05)	**<0.001**

Abbreviations: N: number of patients, SD: standard deviation of mean, Sig: significance level, * based on visceral fat area VFA > 100 cm^2^.

**Table 3 biomedicines-12-02100-t003:** Association of Roseburia and Blautia species with disease characteristics.

Species	Marker	Sig.
*Roseburia hominis*	CDAI	*p* = 0.007 ρ = −0.513
*Roseburia inulinivorans*		*p* = 0.04 ρ = −0.405
*Blautia caecimuris*		*p* = 0.014 ρ = −0.477
*Blautia faecis*		*p* = 0.038 ρ = −0.409
*Blautia massiliensis*		*p* = 0.008 ρ = −0.505
*Blautia wexlerae*		*p* = 0.03 ρ = −0.427
*Blautia obeum*	SES-CD	*p* = 0.04
*Blautia argi*	fecal calprotectin	*p* = 0.018 ρ = −0.471
*Blautia massiliensis*	hematocrite	*p* = 0.02 ρ = 0.452
*Blautia schinkii*	trombocyte	*p* = 0.046 ρ = −0.394
*Roseburia intestinalis*		*p* = 0.034 ρ = −0.418
*Blautia caecimuris*	serum iron	*p* = 0.039 ρ = 0.443
*Blautia faecis*		*p* = 0.018 ρ = 0.498
*Blautia caecimuris*	albumin	*p* = 0.09 ρ = 0.520

Abbreviations: CDAI: Crohn’s disease activity index, SES-CD: Simple Endoscopic Score for Crohn’s disease, ρ: Spearman’s rank correlation coefficient, Sig: significance level.

## Data Availability

The data presented in this study are available on request from the corresponding author. The data are not publicly available due to privacy of individuals that participated in the study.

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
