# Peer review of "Shotgun Analysis of Gut Microbiota with Body Composition and Lipid Characteristics in Crohn’s Disease"

_biomedicines, 2024, doi:10.3390/biomedicines12092100_

Round 1
Reviewer 1 Report
Comments and Suggestions for Authors
The Manuscript entitled “Shotgun Analysis of Gut Microbiota with body composition and 2 Lipid Characteristics in Crohn’s Disease” has been written well. However, it still requires some changes as follows:
All tables in the manuscript should be numbered in chronological order.
Figure 1: The Y-axis should also be increased to capture SD values.
Line 230: Figure 2 is labeled as Figure 1. Please correct it and
Please make all figures of high resolution, with visible x and Y axis values and scales.
The rest manuscript is well-written without any errors.
Thanks
Author Response
- All tables in the manuscript should be numbered in chronological order.
Thank you for your comment. We have corrected the numbering of the tables.
- Figure 1: The Y-axis should also be increased to capture SD values.
We have changed all the figures throughout the manuscript to higher resolution and vector file.
- Line 230: Figure 2 is labeled as Figure 1. Please correct it
The label of the figure has been corrected.
- Please make all figures of high resolution, with visible x and Y axis values and scales.
We have changed all the figures throughout the manuscript to higher resolution and vector file.
Reviewer 2 Report
Comments and Suggestions for Authors
The introduction section is poorly structured and lacks clarity. It does not provide a clear overview of the research problem, objectives, and significance. A well-defined introduction should offer a comprehensive background to set the stage for the research question. Consider revisiting the introduction to present a more focused and coherent background of the study.
The introduction fails to adequately explain the context and relevance of the study. It is essential to highlight the research gap and justify the need for this study within the context of existing literature. This can be improved by citing relevant studies and explaining how this research aims to address any identified gaps.
Certain sections of the paper seem disconnected from the main research theme. The content should be reviewed to ensure that each section contributes directly to the research question and objectives. Avoid including tangential information that does not add value to the study.
The language throughout the paper often lacks precision, leading to ambiguity and confusion.
Line 53: “A decrease in Bacillota has been associated with disease flares in inflammatory bowel disease (IBD) and has been observed in individuals with obesity.” How about in non-obese patients, if there is a bacterial alteration in non-obese patients with IBD?
Line 60: “Several studies revealed that the human gut microbiota is influenced by diet and nutritional habits, and these are particularly evident in obesity.” What is the reference?
Line 73: There is no conclusion at the end of the paragraph
Line 311: “The possible connection of lipid metabolism and body composition associated microbiota alterations in patients with CD was investigated.” is it The possible connection of lipid metabolism and body composition "with" or "and" associated microbiota alterations: this is confusing.
Line 320: “Most patients with IBD "sufferred" from malnutrition during the disease course, which can be measured in 6–70 %. [24] Sarcopenia is associated with increased risk for hospital ad-321 mission, worse clinical outcomes, and poor quality of life. [25] Nevertheless, more and more patients with IBD characterized by obesity are presenting at regular visits. Data suggest that high visceral fat and obesity are associated with increased disease activity, development of extraintestinal symptoms, reduced efficacy of biological treatments, and poor disease outcomes. [26;27]: this is confusing and irrelevant. "sufferred" needs to be corrected to "suffered"
The paper requires significant revision.
Comments on the Quality of English LanguageThe language throughout the paper often lacks precision, leading to ambiguity and confusion.
Author Response
- The introduction section is poorly structured and lacks clarity. It does not provide a clear overview of the research problem, objectives, and significance. A well-defined introduction should offer a comprehensive background to set the stage for the research question. Consider revisiting the introduction to present a more focused and coherent background of the study.
Thank you for your comments. The Introduction section has been revised, reconstructed and simplified to clarify the purpose of our study and show the relationship between IBD, obesity, microbiota and proinflammatory processes.
- The introduction fails to adequately explain the context and relevance of the study. It is essential to highlight the research gap and justify the need for this study within the context of existing literature. This can be improved by citing relevant studies and explaining how this research aims to address any identified gaps.
Thank you. The Introduction section has been revised, reconstructed and simplified. References to relevant studies were included, as well.
- Certain sections of the paper seem disconnected from the main research theme. The content should be reviewed to ensure that each section contributes directly to the research question and objectives. Avoid including tangential information that does not add value to the study.
Thank you for the comment, the manuscript has been simplified and corrected. Although it does not seem to be closely related to our objectives, it is unavoidable to take into account the IBD disease activity and the type of therapy when analysing this topic. We therefore consider it important to include these results in our study.
- The language throughout the paper often lacks precision, leading to ambiguity and confusion.
The manuscript has been revised and the spelling and fluency corrected by a native English speaker.
- Line 53: “A decrease in Bacillota has been associated with disease flares in inflammatory bowel disease (IBD) and has been observed in individuals with obesity.” How about in non-obese patients, if there is a bacterial alteration in non-obese patients with IBD?
Thank you for your valuable comment. The sentence was misleading, since the increased abundance of Bacillota was observed in case of obesity in several cohorts. The sentence was corrected, and a reference was added.
- Line 60: “Several studies revealed that the human gut microbiota is influenced by diet and nutritional habits, and these are particularly evident in obesity.” What is the reference?
The Introduction section has been revised and reconstructed. The connecting reference has been added.
- Line 73: There is no conclusion at the end of the paragraph
The Introduction section has been revised and reconstructed.
- Line 311: “The possible connection of lipid metabolism and body composition associated microbiota alterations in patients with CD was investigated.” is it The possible connection of lipid metabolism and body composition "with" or "and" associated microbiota alterations: this is confusing.
Thank you for your important comment. In this cross-sectional trial, the possible association of microbiota alterations of obesity and IBD and the lipid metabolism and IBD were investigated. Due to the study design and the possible pathogenetic connection of obesity and dysfunction of lipid metabolism (metabolic syndrome), the connection of obesity and lipid metabolism related to microbiota alterations cannot be ruled out. Further studies are needed. The Discussion section has been complemented.
- Line 320: “Most patients with IBD "sufferred" from malnutrition during the disease course, which can be measured in 6–70 %. [24] Sarcopenia is associated with increased risk for hospital ad-321 mission, worse clinical outcomes, and poor quality of life. [25] Nevertheless, more and more patients with IBD characterized by obesity are presenting at regular visits. Data suggest that high visceral fat and obesity are associated with increased disease activity, development of extraintestinal symptoms, reduced efficacy of biological treatments, and poor disease outcomes. [26;27]: this is confusing and irrelevant. "sufferred" needs to be corrected to "suffered"
The paragraph has been revised and corrected. The sarcopenia aspects have been deleted.
Round 2
Reviewer 2 Report
Comments and Suggestions for Authors
Introduction: The current introduction is confusing and covers many unrelated subjects:
Line 49: "Several papers have been reported connections since decrease in Bacillota has been associated with disease flares in inflammatory bowel disease (IBD), while an increased abundance has been observed in individuals with obesity. "======> This is a confusing sentence, what is the connection between Obesity, IBD and gut microbial alteration?
Line 53: “Alterations in the genera Blautia, Roseburia, and Clostridia have been observed in the pathogenesis of pouchitis.”======> It is unrelated
Line 78: "As supported by a recent systematic review and meta‐analysis there is a proven link between certain medications in IBD and hyperlipidemia, but the relationship of hypertriglyceridemia and hypercholesterolemia with the altered IBD gut-microbiota is not exactly understood."======> This sentence is confusing. Which kind of medication? What is the mechanism? What is the connection between medication-related hyperlipidemia in IBD and gut microbiome?
To strengthen the introduction, it is essential to provide a more comprehensive background on the relationship between obesity, hyperlipidemia and the gut microbiome, emphasizing how this connection impacts inflammatory bowel disease (IBD). Begin by discussing the effects of obesity on the gut microbiome, highlighting the reduced microbial diversity, shifts in bacterial populations, and subsequent gut barrier dysfunction. This dysfunction is a key factor leading to low-grade inflammation, which is critical to understanding the broader implications of obesity-related complications. explore the relationship between hyperlipidemia and the gut microbiome, noting how dyslipidemia can lead to gut barrier dysfunction and alterations in gut microbial composition.
Next, bridge the discussion to how these dyslipidemia/ obesity-related changes in the gut microbiome, particularly gut dysbiosis, play a role in IBD. Focus on the mechanisms by which the proinflammatory state associated with obesity, including elevated levels of cytokines like TNF-α and IL-6, contributes to IBD as a chronic inflammatory condition. Address the altered immune regulation seen in obesity, specifically the imbalance between pro-inflammatory and anti-inflammatory responses, and how this imbalance creates an environment that exacerbates IBD symptoms and disease progression. Explain how elevated lipid levels in the blood can contribute to systemic inflammation, which is particularly relevant to the pathogenesis of IBD.
Finally, it is important to connect these concepts to the clinical outcomes in IBD patients, particularly those who are obese. Discuss how obesity not only worsens the severity and progression of IBD but also reduces the effectiveness of treatments. This comprehensive approach will provide the necessary background and context, ensuring that the introduction is both thorough and well-supported by relevant references.
I would suggest considering the removal of discussions related to the role of diet in gut microbiome or IBD treatment from your paper. Since these topics are not the primary focus of your current work, streamlining the content to emphasize your core areas of research might enhance the clarity and impact of your paper. This approach will allow you to concentrate more on the specific aspects of your study that align with your objectives, ensuring a more cohesive and focused presentation.
Line 137: what is the references for : “Visceral fat was measured via body composition analysis and used to categorize patients 137 into obese (above 100 cm2) and non-obese groups.”
Line 140: what is the definition of obesity based on BMI with reference?
Line 247: you only mentioned the WHO definition of obesity but what is the definition itself?
According to WHO obesity is categorized based on BMI score of 30 and more, and BMI between 25 and 29.9 is overweight, NOT Obese as you mentioned the obesity definition in your demographic table (Table 1); the results must be changed based on the correct definition of obesity BMI of 30 and more.
Line 360: “Food intake and diet have significant impact on the intestinal microbiota composition. [39] Food additives, fiber, and carbohydrate ratios define the gut flora, which could be associated with obese phenotype. Therapeutic options involving dietary restrictions in IBD were examined. A low-FODMAP diet had a mild efficacy in improving symptoms [40], while exclusive enteral nutrition is well-established in the pediatric IBD population. [41] In our study participants, 11 out of 15 followed a diet low in fat, fiber, and lactose, and 5 of them had a high BMI, indicating that there are no general dietary therapy recommendations.” How did you conclude there are no general dietary therapy recommendations?
Comments on the Quality of English LanguageThe English is very difficult to understand/incomprehensible.
Author Response
- Line 49: "Several papers have been reported connections since decrease in Bacillota has been associated with disease flares in inflammatory bowel disease (IBD), while an increased abundance has been observed in individuals with obesity. "======> This is a confusing sentence, what is the connection between Obesity, IBD and gut microbial alteration?
The “connection” is referred to diseases which are coupled with altered Bacillota/Bacteroidota ratio. The sentence has been corrected.
- Line 53: “Alterations in the genera Blautia, Roseburia, and Clostridia have been observed in the pathogenesis of pouchitis.”======> It is unrelated
We have simplified the Introduction section and deleted this sentence.
- Line 78: "As supported by a recent systematic review and meta‐analysis there is a proven link between certain medications in IBD and hyperlipidemia, but the relationship of hypertriglyceridemia and hypercholesterolemia with the altered IBD gut-microbiota is not exactly understood."======> This sentence is confusing. Which kind of medication? What is the mechanism? What is the connection between medication-related hyperlipidemia in IBD and gut microbiome?
We have simplified the Introduction section and deleted this sentence.
- To strengthen the introduction, it is essential to provide a more comprehensive background on the relationship between obesity, hyperlipidemia and the gut microbiome, emphasizing how this connection impacts inflammatory bowel disease (IBD). Begin by discussing the effects of obesity on the gut microbiome, highlighting the reduced microbial diversity, shifts in bacterial populations, and subsequent gut barrier dysfunction. This dysfunction is a key factor leading to low-grade inflammation, which is critical to understanding the broader implications of obesity-related complications. explore the relationship between hyperlipidemia and the gut microbiome, noting how dyslipidemia can lead to gut barrier dysfunction and alterations in gut microbial composition.
We have reconstructed and rewrited the Introduction section guided by your suggestions.
- Next, bridge the discussion to how these dyslipidemia/ obesity-related changes in the gut microbiome, particularly gut dysbiosis, play a role in IBD. Focus on the mechanisms by which the proinflammatory state associated with obesity, including elevated levels of cytokines like TNF-α and IL-6, contributes to IBD as a chronic inflammatory condition. Address the altered immune regulation seen in obesity, specifically the imbalance between pro-inflammatory and anti-inflammatory responses, and how this imbalance creates an environment that exacerbates IBD symptoms and disease progression. Explain how elevated lipid levels in the blood can contribute to systemic inflammation, which is particularly relevant to the pathogenesis of IBD.
We have reconstructed and rewrited the Discussion section guided by your suggestions.
- Finally, it is important to connect these concepts to the clinical outcomes in IBD patients, particularly those who are obese. Discuss how obesity not only worsens the severity and progression of IBD but also reduces the effectiveness of treatments. This comprehensive approach will provide the necessary background and context, ensuring that the introduction is both thorough and well-supported by relevant references.
We have reconstructed and rewrited the Discussion section guided by your suggestions.
- I would suggest considering the removal of discussions related to the role of diet in gut microbiome or IBD treatment from your paper. Since these topics are not the primary focus of your current work, streamlining the content to emphasize your core areas of research might enhance the clarity and impact of your paper. This approach will allow you to concentrate more on the specific aspects of your study that align with your objectives, ensuring a more cohesive and focused presentation.
We have removed dietary aspects of obesity, dyslipidaemia and microbiota changes from Discussion.
- Line 137: what is the references for : “Visceral fat was measured via body composition analysis and used to categorize patients 137 into obese (above 100 cm2) and non-obese groups.”
The reference has been added.
- Line 140: what is the definition of obesity based on BMI with reference?
We have analysed high visceral fat by BMI as well (near VFA). BMI above 25 was classified as patients with high visceral fat since according to WHO definition, normal range is between 18,5-25. The obesity definition relates only to VFA. The Methodology, Results and Discussion section has been clarified.
- Line 247: you only mentioned the WHO definition of obesity but what is the definition itself?
The definition of obesity is detailed and complemented in the Definitions section.
- According to WHO obesity is categorized based on BMI score of 30 and more, and BMI between 25 and 29.9 is overweight, NOT Obese as you mentioned the obesity definition in your demographic table (Table 1); the results must be changed based on the correct definition of obesity BMI of 30 and more.
We are absolutely agreeing with your comment. According to the WHO definition BMI above 25 is classified as overweight and obese (above 30) patients. Any increased fat is associated with increased risk of disease, therefore we did not restricted classification to patients only above 30 kg/m2 of BMI, rather above to normal (>25 kg/m2). We have clarified and corrected the BMI and “obesity” definitions to high visceral fat throughout the manuscript.
- Line 360: “Food intake and diet have significant impact on the intestinal microbiota composition. [39] Food additives, fiber, and carbohydrate ratios define the gut flora, which could be associated with obese phenotype. Therapeutic options involving dietary restrictions in IBD were examined. A low-FODMAP diet had a mild efficacy in improving symptoms [40], while exclusive enteral nutrition is well-established in the pediatric IBD population. [41] In our study participants, 11 out of 15 followed a diet low in fat, fiber, and lactose, and 5 of them had a high BMI, indicating that there are no general dietary therapy recommendations.” How did you conclude there are no general dietary therapy recommendations?
This sentence was misleading; therefore, we have deleted the conclusion.
